# White Box Watermarking for Convolution Layers in Fine-Tuning Model Using the Constant Weight Code

**DOI:** 10.3390/jimaging9060117

**Published:** 2023-06-09

**Authors:** Minoru Kuribayashi, Tatsuya Yasui, Asad Malik

**Affiliations:** 1Graduate School of Natural Science and Technology, Okayama University, Okayama 700-8530, Japan; yasui.tatsuya@s.okayama-u.ac.jp; 2Department of Computer Science, Aligarh Muslim University, Aligarh 202001, India; amalik.cs@amu.ac.in

**Keywords:** DNN watermark, fine-tuning model, constant weight code, detection, non-fungible token

## Abstract

Deep neural network (DNN) watermarking is a potential approach for protecting the intellectual property rights of DNN models. Similar to classical watermarking techniques for multimedia content, the requirements for DNN watermarking include capacity, robustness, transparency, and other factors. Studies have focused on robustness against retraining and fine-tuning. However, less important neurons in the DNN model may be pruned. Moreover, although the encoding approach renders DNN watermarking robust against pruning attacks, the watermark is assumed to be embedded only into the fully connected layer in the fine-tuning model. In this study, we extended the method such that the model can be applied to any convolution layer of the DNN model and designed a watermark detector based on a statistical analysis of the extracted weight parameters to evaluate whether the model is watermarked. Using a nonfungible token mitigates the overwriting of the watermark and enables checking when the DNN model with the watermark was created.

## 1. Introduction

Digital watermarking [1,2,3,4] is the technique of inserting messages into multimedia content for various purposes, including copyright protection, authentication, access control, and broadcast monitoring. Depending on the message used as a watermark and the method of recovery of the message from the host signal, watermarking techniques can be categorized into two methods, namely multibit and zero-bit watermarking. In multibit watermarking, the watermark is extracted as the bit string of the message, whereas the presence of a hidden watermark is checked in the case of zero-bit watermarking. Multibit is termed watermark extraction, and zero-bit is termed watermark detection. To extract a watermark, determining the presence of a watermark in a given host signal is critical because even if no watermark is embedded, a certain message is extracted. To prevent incorrect extraction when no watermark is embedded, checking whether watermark information exists in the host signal, that is, watermark detection, is critical.

### 1.1. Background

Improvements in computer performance and the generation of a large amount of data have encouraged the development of deep learning technology. In deep neural networks (DNNs), training the weight parameters requires considerable computational resources and a large amount of data for target application. Expert tuning of several hyperparameters is required for obtaining high performance. Therefore, such a DNN model should be the intellectual property of the owner. The model should be protected against violations of ownership and copyright. A promising approach is to embed a watermark into a DNN model [5]. Originally, watermarking was used for secretly embedding information in multimedia content. In watermark embedding, some forms of redundancy involved in the host signal is exploited such that the perceptual quality of the original content is not degraded. A similar technique is used for DNN watermarking. However, a watermark should be embedded without considerably degrading DNN model performance.

A DNN model consists of several network layers with a large number of weight parameters. Several degrees of freedom exist in the choice of weights. The DNN watermarking technique is based on a slight modification of the weight parameters sampled from many candidates. DNN watermarking is classified into white box and black box techniques based on permissions from the watermark extractor [6]. In the white box technique, internal parameters that directly correspond to the weights of the DNN models are accessible. However, using the black box technique, only the final output of the DNN model is observed. In this case, the watermark is extracted by querying a set of trigger inputs to the DNN model.

In the white box technique, the weight parameters can be accessed not only by the model owner but also by attackers, which renders protecting against external perturbations difficult. Typically, an attacker steals a DNN model and modifies its parameters or structure to fool watermark detection or to cause extraction failure of hidden messages. The primary constraints for the attacker are that the modification of the model should not require more resources than when training the DNN model from scratch, and the performance of the attacked model should not decrease considerably.

An example of perturbation is model pruning in which redundant neurons are pruned without compromising the performance of the DNN model. Initially, pruning is performed to remove less important weight parameters from the DNN model, whose contribution to the loss is small [7]. Attackers can prune some weight parameters to remove the watermark signal. Therefore, robustness against pruning attacks is an important requirement for the white box technique, which ensures that the watermarked parameters are relevant to the original task. In [8], the watermark does not disappear, even after a pruning attack that prunes 65% of the parameters. Li et al. [9] achieved robustness against 60% pruning using the spread transform dither modulation watermarking technique. Zhao et al. [10] embedded watermarks during the pruning process. Tondi et al. [11] proposed a white box multibit watermarking algorithm that adopted the spread spectrum approach [3] to spread watermarks over several weight parameters in a DNN model. Prior to training, the amplitude of such parameters is controlled to be sufficiently large to survive retraining and other modifications.

From the perspective of the communication channel, pruning can be considered to be an erasure channel between the watermark transmitter and receiver. To prevent the erasure of symbols, watermark information is encoded using a binary constant weight code (CWC) [12,13] to ensure robustness against weight-level pruning attacks in [14]. In the encoding method, the symbol “0” is stable even if the corresponding weight parameter are pruned because the symbol is regarded as “0” when it is pruned or its absolute value is small. By producing the bias of symbols “1” and “0” in the code word, high robustness can be achieved against pruning attacks. However, the watermark is embedded only in the fully connected (FC) layer in a fine-tuning model in the experiment. To adapt to any DNN model, evaluating its performance quantitatively when a watermark is embedded in the convolution layer is necessary. Furthermore, although the watermark detection approach has been discussed, a statistical model of the watermarked weight parameters is yet to be theoretically analyzed.

### 1.2. Our Contributions

In this study, we extend the embedding region of the conventional method [14] to any layer of the DNN model. The convolutional neural network consints of multiple layers, including convolution, pooling, dropout, and batch normalization. The suitability of these types of networks for embedding watermarks was evaluated in terms of the transparency and secrecy of the hidden messages and the performance of the original task assigned to the DNN model.

We theoretically analyzed the distribution of the weight parameters and designed a detector to check for the presence of hidden watermarks for the DNN watermarking method [14]. The initial weight parameters of the DNN model were assumed to be random sequences following a uniform distribution, and their distributions were assumed to be approximately equivalent, even after model training. Furthermore, the distribution of the pretrained CNN models was approximately Gaussian. However, the distribution of the weight parameters selected for embedding the watermark was biased according to the CWC code word because the embedding process was performed during DNN model training. In the proposed method, this bias is used to determine whether the selected weight parameter is a CWC code word or a random sequence.

The contributions of this study are summarized as follows:In the DNN model, a suitable layer for embedding the watermark is quantitatively evaluated among multiple layers. Because the number of parameters for convolutional operations is considerably larger than for other operations, the secrecy of the choice of weight parameters can be controlled. Even if these weight parameters are modified by embedding the watermark at the initial setup and are fixed during the training phase, local minima whose model performance is close to other local minima can be determined.Under the assumption that the weight parameters are uniformly distributed or Gaussian, if the watermark is encoded by CWC, the statistical bias of the weight parameters extracted from the watermarked and nonwatermarked DNN models is formulated in the analysis. Therefore, a simple threshold calculated from the statistical distribution can be used to determine the presence of a hidden message.To protect against overwriting attacks, we introduced a nonfungible token (NFT) in the watermark. This token enables us to check the history of the hidden message; the tokenId of the NFT is encoded in the CWC code word and embedded as a watermark.

### 1.3. Organization

The remainder of this paper is organized as follows. In Section 2, we provide an overview of DNN watermarking and its threats. In Section 3, we review the conventional method using the CWC. The proposed method is described in detail in Section 4, and the experimental results are presented in Section 5. Finally, the conclusion to this paper is presented in Section 6, and directions for future research are highlighted.

## 2. Preliminaries

The DNN watermarking is a method of embedding a digital watermark into a DNN model to safeguard intellectual property rights and to prohibit unauthorized use or distribution. This section reviews the numerous DNN watermarking strategies and possible attack scenarios.

### 2.1. DNN Watermarking

Various watermarking techniques have been devised to protect multimedia content, such as audio, still images, video, and text. A watermark signal is inserted into the host signal selected from the multimedia content using a secret key. In the embedding process, the redundancy of the host signal can be mitigated without compromising perceptual quality. This technique can be extended to DNN models. During the training phase, the weight parameters are optimized to minimize the loss function to express the difference between the predicted class labels and the ground truth labels. Because of the large number of weight parameters in a DNN model, many degrees of freedom exist during parameter tuning in the training phase. This phenomenon enabled inserting a watermark without compromising DNN model performance.

Watermarking techniques should control the trade-off requirements of capacity, robustness, and fidelity [5]. For the DNN watermark, fidelity refers to the capability of the watermarked DNN model to accomplish a task.

In [8,15], a binary sequence of watermark information was embedded into the weight parameters of the convolution layer of a DNN model. The weight parameters were updated during each training process such that multiplication with a secret matrix approached the watermark in parallel with the training of the model. Watermarks can be easily modified by overwriting [16].

Generally, the weight parameters of the DNN model are initialized before the training process and refined to reach a single local minimum after a series of epochs. With increased parameters, a DNN model has many local minima with similar performances [17,18]. Based on these characteristics, a watermark is embedded into the initial values of the selected weight parameters, and the selected parameters are not updated during the training phase in the case of the method presented in [19]. As displayed in Figure 1, the watermarking operation moves the initial point in the parameter space according to the secret key. The embedding operation based on the constraint is the initial assignment of weight parameters to a DNN model. The weight change at each epoch is corrected by iteratively performing the operation.

### 2.2. Threats of DNN Watermark

Similar to multimedia watermarking, certain requirements exist for robustness against potential attacks and indicate the possibility of extracting or detecting a watermark from a perturbed version of the host signal. Attacks include signal processing operations such as adding noise, filtering operations, and lossy compression. For DNN watermarking, threats such as fine-tuning, network pruning, and watermark overwriting, which remove watermarks while maintaining the performance of the DNN model, should be considered.

#### 2.2.1. Transfer Learning and Fine-Tuning

Generally, training a DNN model is extremely expensive in terms of computational resources and large training datasets. Therefore, DNN models that have already been trained are used. To adapt the model pretrained for one task to a new task, the pretrained layers were frozen and replaced with new FC layers. A novel DNN model was trained on the new dataset for the trainable layers above the frozen layer.

If a watermark is embedded in an FC layer, then the watermark is completely removed when the fine-tuning model is created because watermarked weight parameters are replaced, and the weight parameters in the unfrozen layers change. Therefore, the watermark must be robust against the retraining of the unfrozen layer during fine-tuning.

#### 2.2.2. Pruning DNN Model

Even if the DNN model is pretrained, considerable computational resources are required to perform the desired task, and removing redundant weight parameters from the DNN model can efficiently reduce the memory space and computation time [20]. Several heuristic pruning methods have been developed to identify unimportant components in DNN models and to retrain pruned models to recover model performance. Therefore, to create robust DNN watermarking, considering the effects of pruning and changes during retraining are critical.

Pruning methods are roughly classified into three categories. The first method is weight-level pruning, which sets less important weights to zero and does not change the network structure. The other two methods are channel- and layer-level pruning, which can change the network structure but require large computations to determine efficient network modifications with limited performance loss. Therefore, in this study, we focused on weight-level pruning in which after training, the DNN model is compressed by cutting off parameters with absolute values smaller than the threshold value to zero. The threshold value was set such that the accuracy of the model was not degraded considerably.

#### 2.2.3. Overwriting

A new watermark can be embedded in a previously watermarked model to overwrite the original watermark. In [5], overwriting was defined as an attack in which an additional watermark is inserted into the model to render the original watermark undetectable.

Even if a watermark is detectable, a DNN model may contain two signals. In such cases, disputes over the ownership of the model may occur. From a third-party perspective, determining the originality of a watermark is difficult.

## 3. DNN Watermarking Robust against Pruning

Yasui et al. [14] investigated a channel-coding approach for protecting DNN watermarking against pruning attacks. The *k*-bit watermark ***b*** is encoded into a binary code word using CWC.

CWC C(α,L) with parameters α and *L* is a set of binary code words of length *L*, all with weight α; CWC has a fixed Hamming weight. Therefore, the code word $c=(c_{0},c_{1},\ldots ,c_{L-1})$, $c_{i}\in \left\{0,1\right\}$ of CWC satisfies the condition that
(1)$$\sum_{i=0}^{L-1}c_{i}=\alpha ,$$
where α denotes a fixed constant. Because of its simplicity, the Schalkwijk’s algorithm [12] is used in [14], which does not restrict the use of other algorithms [13,21,22,23].

The weights corresponding to the symbols “1” become more than a higher threshold, whereas the weights corresponding to symbols “0” become less than a lower threshold. When the watermarked DNN model is pruned, the subsequent weight parameters are pruned because of the low values, which does not affect the judgment of symbol “0” in the code word.

### 3.1. Embedding

At the initialization of the DNN model, *L* weight parameters $w=(w_{0},w_{1},\ldots ,w_{L-1})$ were selected from *N* candidates according to a secret key. Let $T_{0}$ and $T_{1}$ be lower and higher thresholds, respectively. Subsequently, an encoded watermark ***c*** was embedded in ***w*** under the following constraints:If $c_{i}=1$, then $|w_{i}|\;\geq T_{1}$; otherwise, $|w_{i}|\;\leq T_{0}$, where $T_{0}$ and $T_{1}$ are thresholds satisfying $0<T_{0}<T_{1}$.

During the training process of the DNN model, the weight parameters were iteratively updated to converge to a local minimum. In the proposed method, the changes in weights ***w*** selected for embedding ***c*** were controlled only by the restrictions during the training process.

First, a *k*-bit watermark ***b*** was encoded into the code word ***c***. Here, parameters α and *L* should satisfy the following conditions:(2)2k≤Lα=L!α!(L−α)!<2k+1.

During the embedding operation, weight parameters ***w*** selected from the DNN model were modified into ***w*** using the two thresholds, namely $T_{1}$ and $T_{0}$.
(3)$$w_{i}^{†}=\left\{\begin{array}{cc} w_{i} & \left(c_{i}=1\right)\cap \left(\right|w_{i}\left|\geq T_{1}\right)\\ sgn\left(w_{i}\right)\cdot T_{1} & \left(c_{i}=1\right)\cap \left(\right|w_{i}\left|<T_{1}\right)\\ w_{i} & \left(c_{i}=0\right)\cap \left(\right|w_{i}\left|\leq T_{0}\right)\\ sgn\left(w_{i}\right)\cdot T_{0} & \left(c_{i}=0\right)\cap \left(\right|w_{i}\left|>T_{0}\right) \end{array}\right.,$$
where
(4)$$sgn\left(x\right)=\left\{\begin{array}{cc} 1 & x\geq 0\\ -1 & x<0 \end{array}\right.$$

### 3.2. Extraction

The distribution of the selected weights to be embedded is expected to be the same as the distribution of all candidates (Gaussian or uniform distribution). When embedding a watermark, the change in the distribution depends on thresholds $T_{1}$ and $T_{0}$ as well as the length *L* of the encoded watermark.

### 3.3. Design of Detector

If a watermark is embedded in the weight parameters $w^{\prime }$ selected from the DNN model based on the secret key, then top α values and the remaining values follow various distributions. The top α values follow a uniform distribution in the range $\left[T_{1},\delta \right]$ because the weight parameters are modified based on the threshold value $T_{1}$. Figure 2a displays the distribution of values when a watermark is embedded in the weight parameters, where $U(0,T_{0})$ denotes a uniform distribution in the range $\left[0,T_{0}\right]$. However, weight parameters, except for the top α follow a uniform distribution in the range $\left[0,T_{0}\right]$ because they are modified based on $T_{0}$. If the watermark is not embedded in the weight parameters, then it follows the original distribution, that is, a uniform distribution in the range [0,δ], as displayed in Figure 2b.

A watermark detector was designed by focusing on the difference in the distribution of the weight parameters with and without the watermark. The distribution when the watermark was embedded in the weight parameters is displayed in Figure 2a. Therefore, the average value of the top α weight parameters was $(\delta +T_{1})/2$, whereas that of the L−α weight parameters, except for the top α, was $T_{0}/2$. The distribution when no watermark was embedded is displayed in Figure 2b. The average value of the weight parameter is 1/δ. The proposed detector discriminates differences in the distribution of the weight parameter $c^{\prime }$ extracted from the target DNN model by using the variation in the mean value presented in the previous section as an indicator. The variation index was calculated using the mean square error (MSE) as follows:(5)$$MSE=\frac{1}{L}\sum_{i=0}^{L-1}d_{i},$$
where
(6)$$d_{i}=\left\{\begin{array}{cc} {\left(c_{i}^{\prime }-\frac{\delta +T_{1}}{2}\right)}^{2} & \left(0\leq i<\alpha \right)\\ {\left(c_{i}^{\prime }-\frac{T_{0}}{2}\right)}^{2} & \left(\alpha \leq i<L\right) \end{array}\right.$$

If the watermark is embedded in the weight parameters, the value of MSE is smaller; if not, then the value of MSE is larger.

### 3.4. Recovery of Watermark

First, the weight parameters were selected from the same DNN model positions, denoted by $w^{\prime }$. Subsequently, the αth largest element ${\tilde{w}}_{L-\alpha }^{\prime }$ is determined from $w^{\prime }$, and the code word $c^{\prime }$ is constructed as follows:(7)$$c_{i}^{\prime }=\left\{\begin{array}{cc} 1\; & \;if\;|w_{i}^{\prime }|\geq {\tilde{w}}_{L-\alpha }^{\prime }\\ 0\; & \;\mathrm{otherwise} \end{array}\right.,$$
where ${\tilde{w}}^{\prime }=sort(|w^{\prime }\left|\right)$. Finally, watermark $b^{\prime }$ is reconstructed from the code word $c^{\prime }$.

In the operation, the top-α symbols in $w^{\prime }$ are regarded as “1”, and the others are “0”. Even if L−α symbols whose absolute values are smaller than those of the top-α symbols are pruned, the code word can be correctly reconstructed from the weight parameters $w^{\prime }$ in the pruned DNN model. When the pruning rate *R* satisfies the condition
(8)$$R<\frac{L-\alpha }{L}=\bar{R},$$

No statistical errors were observed in the aforementioned extraction methods. Because *L* weight parameters $w^{\prime }$ are sampled from *N* candidates in a DNN model to embed the watermark, the condition does not coincide with the robustness against a pruning attack at rate *R*.

## 4. Proposed DNN Watermarking Method

### 4.1. Embedding Layers

A novel CNN-based model architecture was designed in a fine-tuning model by replacing the last few parts of network layers, including FC layers. Because several common architectures that follow open-source software have been released, Refs. [14,19] considered that a common architecture is not subject to protection. Therefore, the FC layers are targeted for embedding the watermark. However, from the perspective of the intellectual property of the DNN model, common architectures should be protected instead of FC layers.

If an attacker simply makes a copy of a DNN model, the watermark embedded into the fully connected layer will remain in a pirated copy. However, a clever attacker might perform a fine-tuning attack by replacing the fully connected layer with a newly designed one. In such a case, the watermark must be completely removed. Therefore, we can counter the fine-tuning attack by embedding the watermark not only into the fully connected layer but also into the convolution layer.

#### 4.1.1. Characteristics of Convolution Layers

In contrast to conventional studies in [14,19], we extended the network layer targeted for watermarking to any layer of the CNN model. Two factors should be considered in the context of the imperceptibility—one is the secrecy of the choice of weight parameters, and the other is the effect on model performance for the original task.

Although the FC layers are composed of a large number of weight parameters, some networks in the convolution layer, such as batch normalization, pooling, skip connection, and dropout, have a limited number of candidates for selecting parameters. By contrast, convolution networks such as Conv1D and Conv2D have sufficient parameters to be watermarked. For example, the number of parameters in VGG16 [24] is enumerated by Table 1. Therefore, if sufficient candidates exist, embedding a watermark in weight parameters selected from any convolution layer is possible based on a secret key.

In [17,18], a DNN model has many local minima that are almost optimal. Even if some parameters are changed slightly, their influence can be adjusted to be small in the remaining parameters. Thus, if the watermark is embedded in the convolution layer and the watermarked weight parameters are frozen during the retraining phase, the corresponding local minima can be determined by adjusting other weight parameters without sacrificing model performance.

On publication of the pretrained model, a watermark is embedded in the weight parameters selected from the convolutional network according to a secret key, and these parameters are frozen during training. To ensure robustness against retraining attacks, these parameters should be selected from the early stages of the convolutional layer. For fine-tuning, the FC layer should be targeted to embed the watermark because it is a newly designed part of the overall DNN model. The FC layer of the fine-tuning model should be protected because the first common convolution layer should be derived from a model that is publicly available or may be licensed by its owner.

Figure 3 displays the statistical distribution of weight parameters in the three CNN models VGG16 [24], ResNet50 [25], and XceptionNet [26] trained using the ImageNet dataset [27]. The distribution at each layer is close to Gaussian, and the variance decreases with the layer depth.

#### 4.1.2. Design of Threshold

Because of the difference in statistical characteristics, the thresholds $T_{0}$ and $T_{1}$ are determined when the target convolution layer is selected. According to a secret key, we choose *L* weight parameters and modify them by an embedding operation such that the absolute value of α elements is greater than $T_{1}$ and that of the others is less than $T_{0}$.

Let $N_{t}$ be the total number of weight parameters in the *t*-th layer. To avoid statistical irregularities, the absolute value of the top β weight parameters is managed to be greater than $T_{1}$, where β satisfies the following equation:(9)$$\beta =\frac{\alpha }{L}N_{t}=\left(1-\bar{R}\right)N_{t}$$

The weight parameters are $\omega_{t}=\left(\omega_{t,0},\omega_{t,1},\ldots ,\omega_{t,N_{t}-1}\right)$. These parameters are sorted by absolute values such that the order of their absolute values is from largest to smallest, as represented by ${\tilde{\omega }}_{t}$ in the following equation: (10)$${\tilde{\omega }}_{t}=\mathrm{sort}(|\omega_{t}\left|\right)=\mathrm{sort}(|\omega_{t,0}\left|,\right|\omega_{t,1}\left|,\ldots ,\right|\omega_{t,N_{t}-1}\left|\right).$$

Here, ${\tilde{\omega }}_{t,i}>{\tilde{\omega }}_{t,j}>0$ for i<j. Then, threshold $T_{1}$ is determined as follows:(11)$$T_{1}={\tilde{\omega }}_{t,\beta }.$$

Figure 4 illustrates the procedure for determining the threshold $T_{1}$. The other threshold $T_{0}$ is calculated using rate γ:(12)$$T_{0}=\gamma T_{1}$$

For simplicity, γ=1/2 is used in the following discussion.

From the secrecy perspective, attackers can calculate the threshold $T_{1}$ by observing the target convolution layer. However, without a secret key, determining the layer to be selected and identifying the elements that satisfy $|w_{t,j}|\;\geq T_{1}$ is difficult. Because the number of such weight parameters is β, finding α among them renders maintaining the model performance difficult. Another approach is to increase the values of selected weight parameters whose values are $|w_{t,j}|\;\leq T_{0}$. Because of the large number of candidates, finding such weights without a secret key is challenging.

### 4.2. Detector

In the DNN watermarking method [14] explained in the previous section, the bias of 0 and 1 symbols in a CWC code word is exploited to check the presence of hidden watermarks in a DNN model. However, a theoretical analysis is yet to be performed on detecting whether a watermark is embedded in the weight parameters selected from the DNN model based on the secret key. In this study, we theoretically explained the characteristics of the distribution of weight parameters with and without embedded watermarks and proposed a detector that can detect the presence of watermarks.

#### 4.2.1. Measurement

According to the secret key, *L* weight parameters are extracted from the target DNN model. The embedding operations are given by Equation (Equation 3). The upper α values of the weight parameters were equal to or greater than $T_{1}$. Furthermore, when the symbol “0” of the CWC code word is embedded, the weight parameters are modified to be less than or equal to $T_{0}$, and the number of such symbols are L−α. If the watermark is embedded in the selected *L* weight parameters, then their values are not expected to be in the range $\left[T_{0},T_{1}\right]$. Therefore, we proposed a novel measurement method for detecting watermarks.

For detection, we used the same secret key to select *L* weight parameters from the specified convolution layer. Here, thresholds $T_{0}$ and $T_{1}$ are calculated from the selected weight parameters by observing only the distribution with parameter α of the CWC code word and rate γ. Here, $T_{1}$ is determined using the weight parameter with the absolute value of the αth largest, and $T_{0}$ is calculated using Equation (Equation 12).

The following figure illustrates how the watermark is detected using the proposed method.

Extract *L* weight parameters $c^{\prime }$ based on the secret key from the DNN model.Sort $c^{\prime }$ in descending order:
(13)$$\mathrm{sort}\left(c^{\prime }\right)=\tilde{c}=\left({\tilde{c}}_{0},\ldots ,{\tilde{c}}_{\alpha -1},{\tilde{c}}_{\alpha },\ldots ,{\tilde{c}}_{\eta -1},{\tilde{c}}_{\eta },\ldots ,{\tilde{c}}_{L-1}\right),$$
where $T_{1}={\tilde{c}}_{\alpha }$, $T_{0}=\gamma T_{1}$, and
(14)$$\eta =\underset{0\leq i<L}{\arg\;\min}\left[{\tilde{c}}_{i}>T_{0}\right].$$Calculate the modified $\tilde{MSE}$ from $\tilde{c}$ satisfying ${\tilde{c}}_{i}>T_{0}$, except for the top α parameters.
(15)$$\tilde{MSE}=\frac{1}{\eta -\alpha +1}\sum_{i=\alpha }^{\eta }{\left({\tilde{c}}_{i}-T_{0}\right)}^{2}$$Determine the presence of the watermark if the $\tilde{MSE}$ value exceeds a detection threshold.

If the watermark is not embedded, then the sorted weight parameters $\tilde{c}$ should satisfy the following conditions:(16)$$\left\{\begin{array}{cc} {\tilde{c}}_{i}\geq T_{1} & \left(0\leq i<\alpha \right)\\ T_{0}<{\tilde{c}}_{i}<T_{1} & \left(\alpha \leq i\leq \eta \right)\\ {\tilde{c}}_{i}\leq T_{0} & \left(\eta <i<L\right) \end{array}\right.$$

During the embedding process, for α≤i<η, ${\tilde{c}}_{i}$ is maintained at less than $T_{0}$. Therefore, $\tilde{MSE}$ should be close to 0 if the watermark is to be embedded in a DNN model.

#### 4.2.2. Uniform Distribution

The initial weight parameters in a DNN model are assumed to be uniformly distributed in the range [−δ,δ], and these absolute values are considered for simplicity.

As displayed in Figure 2a, if the watermark is embedded, then the probability in the range $\left[T_{0},T_{1}\right]$ should be zero. The theoretical expected value of $\tilde{MSE}$ can be obtained using the following formula:**CWC codword:** ${\tilde{E}}_{C}^{\mathrm{unif}}$
(17)$${\tilde{E}}_{C}^{\mathrm{unif}}=\int_{T_{0}}^{T_{1}}{\left(x-T_{0}\right)}^{2}\cdot 0dx=0$$
**Random sequence:** ${\tilde{E}}_{N}^{\mathrm{unif}}$
(18)$${\tilde{E}}_{N}^{\mathrm{unif}}=\int_{T_{0}}^{T_{1}}{\left(x-T_{0}\right)}^{2}\cdot \frac{1}{2\delta }dx+\int_{-T_{1}}^{-T_{0}}{\left(x+T_{0}\right)}^{2}\cdot \frac{1}{2\delta }dx=\frac{1}{\delta }\int_{0}^{T_{1}-T_{0}}x^{2}dx=\frac{1}{3\delta }{\left(T_{1}-T_{0}\right)}^{3}$$

If $T_{0}=T_{1}/2$ (i.e., γ=1/2), then ${\tilde{E}}_{N}^{\mathrm{unif}}=T_{1}^{3}/24\delta$.

#### 4.2.3. Gaussian Distribution

After training, the weight parameters in the DNN model were expected to follow a Gaussian distribution with a mean of zero and variance $\sigma^{2}$. For simplicity, we considered the absolute values.

**CWC code word:** 

${\tilde{E}}_{C}^{\mathrm{Gauss}}$




(19)
$${\tilde{E}}_{C}^{\mathrm{Gauss}}=\int_{T_{0}}^{T_{1}}{\left(x-T_{0}\right)}^{2}\cdot 0dx=0$$


**Random sequence:** 

${\tilde{E}}_{N}^{\mathrm{Gauss}}$




(20)
$$\begin{array}{cc} {\tilde{E}}_{N}^{\mathrm{Gauss}}= & \int_{T_{0}}^{T_{1}}{\left(x-T_{0}\right)}^{2}\cdot \frac{1}{\sqrt{2\pi \sigma^{2}}}e^{-\frac{x^{2}}{2\sigma^{2}}}dx+\int_{-T_{1}}^{-T_{0}}{\left(x+T_{0}\right)}^{2}\cdot \frac{1}{\sqrt{2\pi \sigma^{2}}}e^{-\frac{x^{2}}{2\sigma^{2}}}dx\\ = & \frac{2}{\sqrt{\pi }}\int_{\frac{T_{0}}{\sqrt{2}\sigma }}^{\frac{T_{1}}{\sqrt{2}\sigma }}\left(2\sigma^{2}x^{2}-2\sqrt{2}\sigma T_{0}x+T_{0}^{2}\right)e^{-x^{2}}dx\\ = & \frac{4\sigma^{2}}{\sqrt{\pi }}\int_{\frac{T_{0}}{\sqrt{2}\sigma }}^{\frac{T_{1}}{\sqrt{2}\sigma }}x^{2}e^{-x^{2}}dx-\frac{4\sqrt{2}\sigma T_{0}}{\sqrt{\pi }}\int_{\frac{T_{0}}{\sqrt{2}\sigma }}^{\frac{T_{1}}{\sqrt{2}\sigma }}xe^{-x^{2}}dx+\frac{2T_{0}^{2}}{\sqrt{\pi }}\int_{\frac{T_{0}}{\sqrt{2}\sigma }}^{\frac{T_{1}}{\sqrt{2}\sigma }}e^{-x^{2}}dx\\ = & \frac{4\sigma^{2}}{\sqrt{\pi }}\left(-\frac{T_{1}}{2\sqrt{2}\sigma }e^{-\frac{T_{1}^{2}}{2\sigma^{2}}}+\frac{T_{0}}{2\sqrt{2}\sigma }e^{-\frac{T_{0}^{2}}{2\sigma^{2}}}-\frac{1}{2}\int_{\frac{T_{0}}{\sqrt{2}\sigma }}^{\frac{T_{1}}{\sqrt{2}\sigma }}-e^{-x^{2}}dx\right)\\ \; & \;-\frac{4\sqrt{2}\sigma T_{0}}{\sqrt{\pi }}\left(-\frac{1}{2}e^{-\frac{T_{1}^{2}}{2\sigma^{2}}}+\frac{1}{2}e^{-\frac{T_{0}^{2}}{2\sigma^{2}}}\right)+\frac{2T_{0}^{2}}{\sqrt{\pi }}\int_{\frac{T_{0}}{\sqrt{2}\sigma }}^{\frac{T_{1}}{\sqrt{2}\sigma }}e^{-x^{2}}dx\\ = & -\frac{\sqrt{2}\sigma T_{0}}{\sqrt{\pi }}e^{-\frac{T_{0}^{2}}{2\sigma^{2}}}+\frac{\sqrt{2}\sigma }{\sqrt{\pi }}\left(2T_{0}-T_{1}\right)e^{-\frac{T_{1}^{2}}{2\sigma^{2}}}\\ \; & \;+\left(\sigma^{2}+T_{0}^{2}\right)\left(\mathrm{erfc}\left(\frac{T_{0}}{\sqrt{2}\sigma }\right)-\mathrm{erfc}\left(\frac{T_{1}}{\sqrt{2}\sigma }\right)\right) \end{array}$$


If $T_{0}=T_{1}/2$, then we have the following equation:
(21)$${\tilde{E}}_{N}^{\mathrm{Gauss}}=\left(\sigma^{2}+\frac{T_{1}^{2}}{4}\right)\left(\mathrm{erfc}\left(\frac{T_{1}}{2\sqrt{2}\sigma }\right)-\mathrm{erfc}\left(\frac{T_{1}}{\sqrt{2}\sigma }\right)\right)-\frac{\sqrt{2}\sigma T_{1}}{2\sqrt{\pi }}e^{-\frac{T_{1}^{2}}{8\sigma^{2}}},$$
where erfc() denotes the complementary error function defined by the following equation:
(22)$$\mathrm{erfc}\left(a\right)=\frac{2}{\sqrt{\pi }}\int_{a}^{\infty }e^{-x^{2}}dx$$

### 4.3. Non-Fungible Token

Even if a malicious party attempts to overwrite the watermark, then removing the hidden watermark from a watermarked DNN model without a secret key is difficult. When a new watermarked version is created using a different key and watermark, the two watermarks must remain. An nonfungible token (NFT) is introduced to check the operation history of the watermark.

Since the invention of Bitcoin [28], blockchain has been adopted in various new guaranteed applications based on a consensus algorithm called proof of work (PoW). Bitcoin uses the PoW algorithm to reach an agreement on transaction data in a decentralized network.

When the shared data on the blockchain are confirmed by the most distributed nodes, any changes to the stored data are immutable because all subsequent data become invalid. The most popular blockchain platform for NFT schemes is Ethereum [29], which provides a secure environment for running smart contracts. In Blockchain-based smart contracts, a turing-complete scripting language is used for complex functionality to perform a thorough replication of state transitions using a consensus algorithm to achieve ultimate consistency. The applications running on top of smart contracts are based on state-transition mechanisms. The state, including instructions and parameters, is shared by all participants, which guarantees transparency in instruction execution. Furthermore, the positions between states should be the same across the distributed nodes; therefore, consistency is critical.

The NFT system [30] is a blockchain-based application that relies on smart contracts to ensure order-preserving operations. This operation implies that the order of contracts in a blockchain-based application is guaranteed when it is confirmed in the state transitions. Thus, the history of the NFT remains unchanged and ownership is preserved.

A contract address is a unique identifier that comprises a fixed number of alphanumeric characters generated from a pair of public and private keys. To transfer NFTs, the owner sends a transaction to involve smart contracts in the ERC-777 standard (https://ethereum.org/ja/developers/docs/standards/tokens/erc-777/ accessed on 1 May 2023). Among the token standards related to NFTs, ERC-721 (https://eips.ethereum.org/EIPS/eip-721 accessed on 1 May 2023) introduces a NFT standard, in which every NFT has a uint256 variable called tokenId, and the pair of contract addresses and uint256 tokenIds is globally unique. Thus, tokenId can be represented as a 256-bits binary sequence.

Because tokenId can be used as an input to generate special identifications, it can be connected to intellectual multimedia content and DNN models to be protected from unauthorized copying. When a DNN model is combined with NFT, the corresponding tokenId is embedded as a watermark in our method. If two watermarks are extracted, then the watermark history can be checked by evaluating the NFT corresponding to the takenIds. The main purpose of NFT is to guarantee the validity of a DNN model and its timing registered at a trusted center. If the tokenId is embedded as a watermark, its validity can be easily verified.

## 5. Simulations

We encoded a watermark using CWC and then embedded the code word into DNN models to evaluate its effects on DNN models.

### 5.1. Experimental Conditions

Because the bit length of the NFT tokenId is 256, the number of watermarks is fixed at k=256 bits in the experiments. Randomly generated 20 binary sequences of length 256 were encoded by the CWC, and the code words were embedded as a watermark to measure the performance in terms of accuracy and loss. According to [14], the CWC parameters were selected, as listed in Table 2. We calculated the threshold $T_{1}$ according to the method presented in Section 4.1.2 and fixed the other threshold $T_{0}=T_{1}/2$ at the embedding operation. These thresholds were calculated as explained in Section 4.2.1 at detection.

Based on the VGG16 [24], ResNet50 [25], and XceptionNet models as pretrained models, we fine-tuned the models with a batch size of 32 by replacing the new FC layers connected to the final convolutional layer, similar to the experiments in [14,19]. The number of nodes in the final convolutional layer is 25,088 (=7 × 7 × 512) in VGG16, 100,352 (=7  × 7 × 2048) in ResNet50, and 204,800 (=10  × 10 × 2048) in XceptionNet. These nodes are connected to new FC layers with 256 nodes. These fine-tuned models are trained using the 17 Category Flower Dataset (https://www.robots.ox.ac.uk/~vgg/data/flowers/17/) (accessed on 1 May 2023) provided by the Visual Geometry Group of Oxford University—62.5% of images were used as training data, 12.5% were used as validation data, and 25.0% were used as test data. We used the stochastic gradient descent optimizer with a learning rate of $10^{-3}$ and categorical cross-entropy as loss functions in the experiments. Notably, embedding loss was not used in the proposed method. The baseline performances of the fine-tuned model for these three pretrained models are listed in Table 3.

### 5.2. Dependency of Embedding Layer

To evaluate the sensitivities of model performance to watermark embedding, we conducted experiments by embedding watermarks into nine individual convolution layers, where the selected layers and their number of weight parameters are listed in Table 4. In the experiments, we embedded watermark into only one specified convolution layer and compared its performance.

Figure 5 displays the accuracy and loss when the watermark is independently embedded in nine convolution layers in the three DNN models. Compared with the baseline results, no clear difference was observed in performance when the watermark was embedded into the convolution layers in these DNN models. For XceptionNet, a slight decrease in the accuracy was observed when the early stages of the convolutional layers were changed. Embedding a watermark in the first few layers should be avoided. Notably, the loss function does not consider the effect of watermark embedding on the training process.

Next, we compared the performances by changing the CWC encoding parameters, the results of which are displayed in Figure 6. The figure reveals that the accuracy was close to that of the original model in all the cases, although the loss was slightly higher in the case of ResNet50. The difference was within the margin of error when the length of the code word was changed.

Because many candidates exist for the convolution layers, the selection of weight parameters can be kept confidential. If the early convolution layers are frozen during fine-tuning, the embedded watermark can be correctly extracted. As mentioned in Section 4.1.2, the secrecy of the weight parameters to be watermarked is managed by appropriately selecting threshold $T_{1}$. Although we used $T_{0}=T_{1}/2$ in the experiments, it can be flexibly changed to control the secrecy and robustness of the watermark. With a decrease in $T_{0}$, the robustness against the addition of noise increases. Because the watermark weight parameter corresponding to symbol 0 is less than $T_{0}$, the noise amplitude should be greater than $T_{1}-T_{0}$ to cause a bit flip at the symbol position in the code word. By contrast, the performance of DNN model may be degraded because most parameters have small values, similar to pruning.

All code words embedded into DNN models were correctly extracted even if pruning with a maximum rate of 99% was executed because of the effect of encoding by CWC at a rate $\bar{R}$. Even if the pruned models were retrained with a few epochs, the code words still survived because the changes in the weight parameters owing to retention were considerably smaller than the gap $T_{1}-T_{0}$ in the experiments.

### 5.3. Detection

The modified $\tilde{MSE}$ was measured for the fine-tuned model after training. When the watermark was embedded in the model, $\tilde{MSE}=0$ in all cases. However, this model varied for the layers in which the watermark was embedded. Figure 7 displays the experimental $\tilde{MSE}$ and theoretical values ${\tilde{E}}_{C}^{\mathrm{Gauss}}$ for each layer, where the standard derivation σ was calculated from *L* selected weight parameters. The experimental values were slightly lower than the theoretical values because the weight parameters were biased away from a Gaussian distribution. However, the difference between the values of the weight parameters with and without the watermark enables the detection of the presence of the watermark according to theoretical values.

### 5.4. Comparison

We compared the proposed method with a white box DNN watermarking method [8,9,11,14,31,32,33]. Considering their robustness against transfer learning and fine-tuning, FC layers should be avoided for embedding watermarks because they are replaced by newly designed ones. With the exception of the method described in [14], conventional methods use an embedding loss function to control the distortion of the watermarked model. Therefore, more training epochs were required to converge the watermarked model for the target task. To ensure robustness against pruning attacks, the proposed method can manage a tolerable pruning rate $\bar{R}$ by selecting α and *L* for CWC. The summary of the comparison is enumerated in Table 5.

If an attacker attempts to embed a new watermark, then it is likely to overwrite and remove existing watermarks at these bit positions. Conventional methods consider robustness against such attacks by rendering the extraction operation complex and guessing the secret key difficult. Even if the original watermark remains after overwriting, originality cannot be ensured because two individual watermarks are contained in the DNN model. In the proposed method, by introducing NFT to generate the watermark, the time sequence of the watermark and watermark models can be checked.

Similar to the other DNN watermarking methods, there is a trade-off among robustness, capacity, and transparency. If we increase the capacity with the same level of robustness, the transparency is decreased. To increase the capacity, the number of symbols “1” or its code length in a code word must be increased. This results in an increase in bias in the weight parameters in a DNN model, and hence, it will sacrifice the transparency of the hidden watermark.

## 6. Conclusions

In this study, we investigated the sensitivity of the performance of fine-tuning models when watermarks were embedded in various convolutional layers and trained fine-tuned models with the watermark-embedded layer frozen. Because of the differences in the variance of weight parameters selected from each layer, the thresholds for embedding the watermark were determined using statistical analyses. Even if the loss function did not consider the effects of embedding the watermark, both the accuracy and loss of the watermarked model converged, which is similar to the original model, and no significant degradation in performance was observed in the experiments.

Assuming a Gaussian or uniform distribution of the weight parameters, we can estimate the theoretical value of the measurement $\tilde{MSE}$. By analyzing the statistical distribution of selected weight parameters, the presence of a watermark can be detected using the watermark model. From the theoretical value of $\tilde{MSE}$, an appropriate threshold can be determined to detect a watermark.

## Figures and Tables

**Figure 1 jimaging-09-00117-f001:**
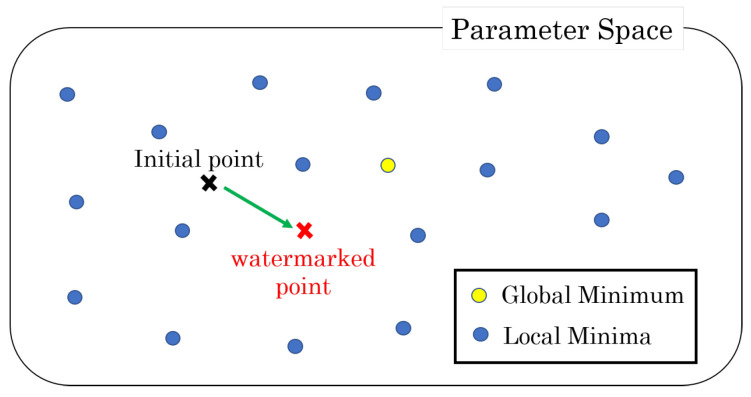
Parameter space in a DNN model. The closest local minimum is selected in a training phase from a given initial point.

**Figure 2 jimaging-09-00117-f002:**
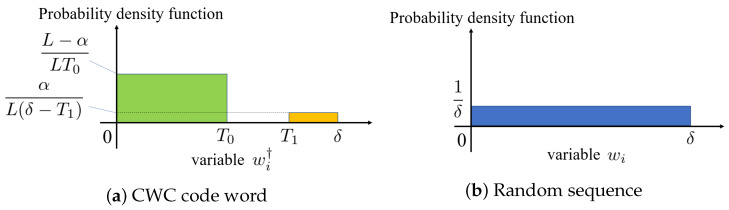
Probability density function of selected weight parameters.

**Figure 3 jimaging-09-00117-f003:**
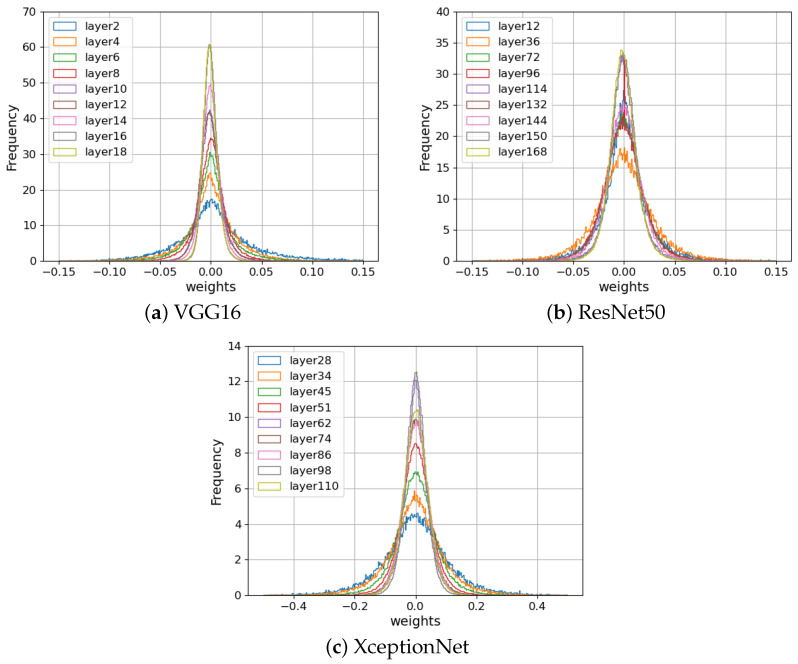
Comparison of the histogram of weight parameters in various convolution layers in pretrained CNN models.

**Figure 4 jimaging-09-00117-f004:**
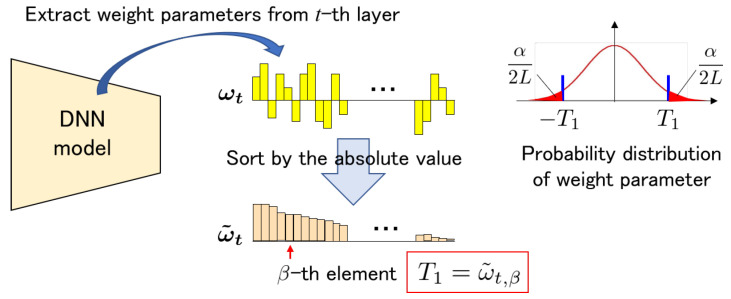
Procedure of calculating a threshold $T_{1}$ from the *t*-th convolution layer of a DNN model.

**Figure 5 jimaging-09-00117-f005:**
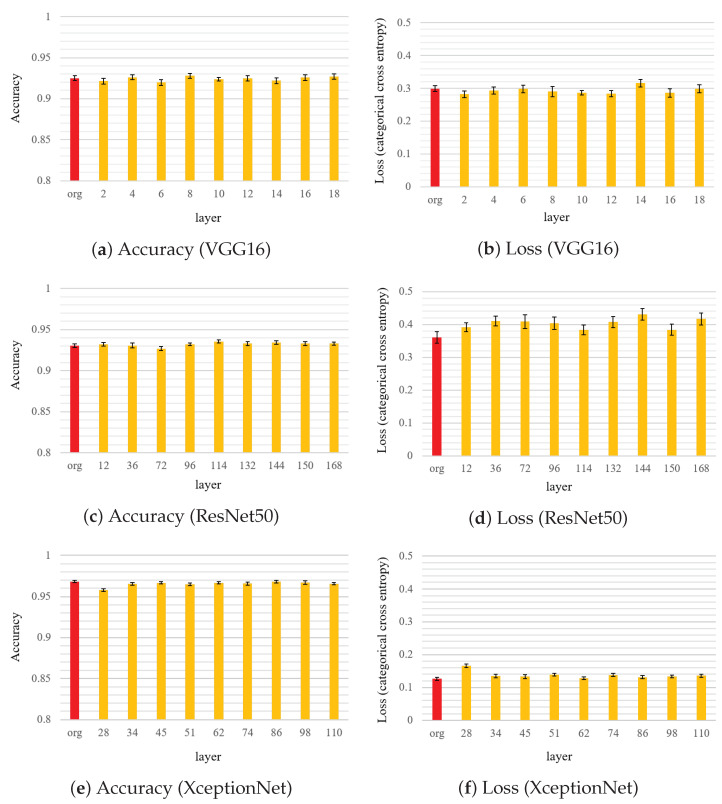
Comparison of accuracy and loss when watermark encoded by CWC(32, 3307) is embedded into each layer, where “org” denotes the original fine-tuning model.

**Figure 6 jimaging-09-00117-f006:**
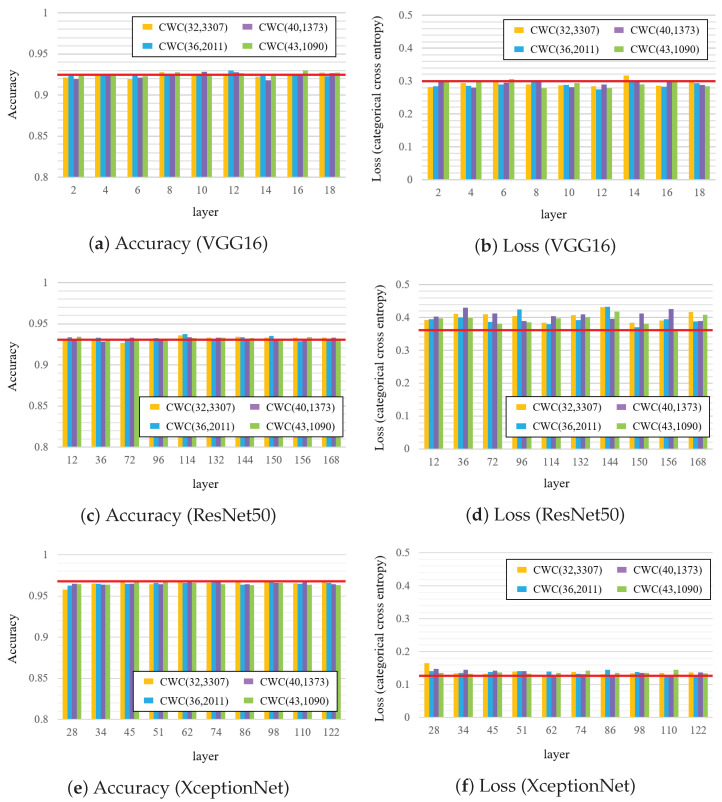
Comparison of accuracy and loss when watermark is embedded into each layer, where the red line denotes the case of original fine-tuning model.

**Figure 7 jimaging-09-00117-f007:**
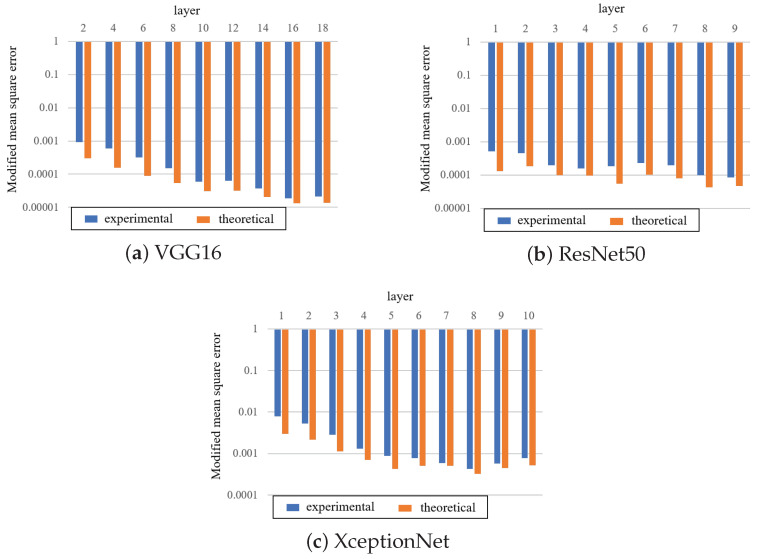
Comparison of $\tilde{MSE}$ in various convolution layers, where watermark is encoded by CWC(32, 3307).

**Table 1 jimaging-09-00117-t001:** Example of weight parameters of a pretrained model for VGG16, which are derived by model.get_weights in the Keras environment.

Layer	Name	Shape	#Param.
0	block1_conv1/kernel:0	(3, 3, 3, 64)	1728
1	block1_conv1/bias:0	(64,)	64
2	block1_conv2/kernel:0	(3, 3, 64, 64)	36,864
3	block1_conv2/bias:0	(64,)	64
4	block2_conv1/kernel:0	(3, 3, 64, 128)	73,728
5	block2_conv1/bias:0	(128,)	128
6	block2_conv2/kernel:0	(3, 3, 128, 128)	147,456
7	block2_conv2/bias:0	(128,)	128
8	block3_conv1/kernel:0	(3, 3, 128, 256)	294,912
9	block3_conv1/bias:0	(256,)	256
10	block3_conv2/kernel:0	(3, 3, 256, 256)	589,824
11	block3_conv2/bias:0	(256,)	256
12	block3_conv3/kernel:0	(3, 3, 256, 256)	589,824
13	block3_conv3/bias:0	(256,)	256
14	block4_conv1/kernel:0	(3, 3, 256, 512)	1,179,648
15	block4_conv1/bias:0	(512,)	512
16	block4_conv2/kernel:0	(3, 3, 512, 512)	2,359,296
17	block4_conv2/bias:0	(512,)	512
18	block4_conv3/kernel:0	(3, 3, 512, 512)	2,359,296
19	block4_conv3/bias:0	(512,)	512
20	block5_conv1/kernel:0	(3, 3, 512, 512)	2,359,296
21	block5_conv1/bias:0	(512,)	512
22	block5_conv2/kernel:0	(3, 3, 512, 512)	2,359,296
23	block5_conv2/bias:0	(512,)	512
24	block5_conv3/kernel:0	(3, 3, 512, 512)	2,359,296
25	block5_conv3/bias:0	(512,)	512
26	dense/kernel:0	(25,088, 256)	6,422,528
27	dense/bias:0	(256,)	256
28	dense_1/kernel:0	(256, 17)	4352
29	dense_1/bias:0	(17,)	17

**Table 2 jimaging-09-00117-t002:** CWC parameters CWC(α,L) for k=256, where α is the number of symbol 1 in the code word of length *L*.

*k*	α	*L*	$\bar{R}$
256	32	3307	0.9903
	36	2011	0.9821
	40	1373	0.9709
	43	1090	0.9606

**Table 3 jimaging-09-00117-t003:** Baseline performance of the original fine-tuning model, where the top *t* layers are frozen in the training.

	VGG16	ResNet50	XceptionNet
Frozen layer	15	150	80
Epoch	50	100	30
Accuracy	0.925	0.934	0.970
Loss	0.299	0.361	0.124

**Table 4 jimaging-09-00117-t004:** Number of weight parameters involved in each selected layer.

(a) VGG16									
Layer	2	4	6	8	10	12	14	16	18
#Param.	36,864	73,728	147,456	294,912	589,824	589,824	1,179,648	2,359,296	2,359,296
**(b) ResNet50**									
Layer	12	36	72	96	114	132	144	150	168
#Param.	36,864	36,864	147,456	147,456	147,456	147,456	131,072	589,824	262,144
**(c) XceptionNet**									
Layer	28	34	45	51	62	74	86	98	110
#Param.	32,768	65,536	186,368	529,984	529,984	529,984	529,984	529,984	529,984

**Table 5 jimaging-09-00117-t005:** Comparison with conventional methods.

	Embedding	Embedding	Fine-	Pruning	Overwriting
	Layer	Loss	Tuning		
Uchida et al. [8]	Conv.	Need	∘	<65%	×
Li et al. [9]	Conv.	Need	∘	< 60%	×
Wang et al. [31]	MLP/Conv.	Need	∘	<90%	×
Liu et al. [32]	Conv.	Need	∘	<75%	△
Wang et al. [33]	Conv.	Need	∘	<95%	△
Tondi et al. [11]	Conv.	Need	∘	<90%	×
Yasui et al. [14]	FC	−	∘	<$\bar{R}$	△
Proposed	Conv.	−	∘	<$\bar{R}$	∘

## Data Availability

Data sharing is not applicable to this article.

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
