# Peer review of "White Box Watermarking for Convolution Layers in Fine-Tuning Model Using the Constant Weight Code"

_2313-433X, 2023, doi:10.3390/jimaging9060117_

Round 1

Reviewer 1 Report

A Deep neural network (DNN) watermarking approach is proposed for protecting the intellectual property rights of DNN models.

The method sounds technical and original.

However, I'd like authors to address the following questions:

Justify that using a nonfungible token is a practical approach for most of the systems to be implemented.

Why do we need to extend the fully connected layer in the fine-tuning model to the convolution layer of the DNN model when the fully connected layer works well?

While this work focuses on robustness, how would this work performs in terms of capacity, and transparency?

Author Response

1. Justify that using a nonfungible token is a practical approach for most of the systems to be implemented.

Thank you for your comment.

The main purpose of NFT is to guarantee the validity of a DNN model and it timing registered at a trusted center. If the token ID is embedded as a watermark, its validity can be easily verified.

2. Why do we need to extend the fully connected layer in the fine-tuning model to the convolution layer of the DNN model when the fully connected layer works well?

It is because of the robustness against the fine-tuning attack.

If an attacker simply makes a copy of a DNN model, the watermark embedded into the fully connected layer will remain in a pirated copy. However, a clever attacker might perform a fine-tuning attack by replacing the fully connected layer with a newly designed one. In such a case, the watermark must be completely removed. Therefore, we can counter the fine-tuning attack by embedding the watermark not only into the fully connected layer, but also into the convolution layer.

3. While this work focuses on robustness, how would this work perform in terms of capacity, and transparency?

Thank you for your comment.

Since this study assumes the NFT token ID is expressed in 256 bits, a discussion of capacity is omitted. Similar to the other DNN watermarking methods, there is a trade-off among robustness, capacity, and transparency. If we increase the capacity with the same level of robustness, the transparency is decreased. To increase the capacity, the number of symbols “1” or its code length in a codeword must be increased. This results in the increase of bias in the weight parameters in a DNN model, and hence, it will sacrifice the transparency of the hidden watermark.

Reviewer 2 Report

The paper presents a white box watermarking method for Deep Neural Networks. The paper is well presented, technical discussion is sound, experimental design and results are sufficiently clear and good.

I just have a couple of comments. Firstly, it is not clear to me if in the experiments the watermark is inserted in all convolutional layers, in a subset of them, or in just the mentioned convolutional layer.

Secondly, the idea of identifying the inserted watermark with a blockchain id (NFT) is interesting, but it is not clear what happens when an attacker override the watermark with a method that does not make use of NFTs.

The paper is easily readable, very few typos.

Author Response

1. Firstly, it is not clear to me if in the experiments the watermark is inserted in all convolutional layers, in a subset of them, or in just the mentioned convolutional layer.

In the experiments, we embedded watermark into only one specified convolution layer and compared its performance. To clarify this point, we add the sentence in Section 5.2.

Secondly, the idea of identifying the inserted watermark with a blockchain id (NFT) is interesting, but it is not clear what happens when an attacker override the watermark with a method that does not make use of NFTs.

Thank you for your comment.

As explained in Section 2.2.3, there is a dispute over the ownership of the model. Due to the huge number of candidate parameters in a DNN model, an additional watermark can be embedded without disturbing the extraction of original watermark. This means that two watermarks will be contained after the overwriting attack. In such a case, a third-party cannot determine which watermark is original because the embedding timing is unclear. If NFT is used, the timing registered at a trusted center can be verified. If two watermarks are extracted, then the watermark history can be checked by evaluating the NFT corresponding to the takenIds.